# Association between objectively measured walking steps and sleep in community-dwelling older adults: A prospective cohort study

**Noriyuki Kimura** ◎*, **Yasuhiro Aso**◎, **Kenichi Yabuuchi**◎, **Etsuro Matsubara**◎

Department of Neurology, Faculty of Medicine, Oita University, Oita, Japan

◎ These authors contributed equally to this work.
* noriyuki@oita-u.ac.jp

## Abstract

Physical inactivity and sleep disturbances are major problems in an ageing society. There is increasing evidence that physical activity is associated with sleep quality. However, the association between daily walking steps and sleep remain unclear. This prospective study examined the relationship between objectively measured daily walking steps and sleep parameters in Japanese community-dwelling older adults. In total, 855 community-dwelling individuals aged 65 and above, with an uninterrupted follow-up from August 2015 to March 2016, were enrolled. The participants wore a wristband sensor for an average of 7.8 days every three months. Multiple linear regression analysis was performed to examine the relationship between daily walking steps and sleep parameters, including the total sleep time, sleep efficiency, time awake after sleep onset (WASO), awakening time count during the night, and naptime. The median (interquartile range, IQR) age of the participants was 73 (69–78) years, with 317 (37.1%) men and 538 (62.9%) women. The median (IQR) educational level was 12 (11–12) years, and the median (IQR) Mini-Mental State Examination score was 29 (27–30) points. The number of daily walking steps showed a positive correlation with sleep efficiency and an inverse correlation with WASO, awakening time count, and naptime, after adjusting for covariates and correcting for the false discovery rate ($\beta$ = 0.098, 95% confidence interval [CI]: 0.034 to 0.162, $p$ = 0.003; $\beta$ = −0.107, 95% CI: −0.172 to −0.043, $p$ = 0.001; $\beta$ = −0.105, 95% CI: −0.17 to −0.04, $p$ = 0.002; and $\beta$ = −0.31, 95% CI: −0.371 to −0.249, $p$ < 0.001, respectively). Our results can help promote walking as an intervention for preventing sleep disturbances in community-dwelling older adults.

## Introduction

Sleep disturbances are a major problem in an aging society with an increasing life expectancy [1, 2]. Generally, the prevalence of sleep disorders increases with age. Up to 50% of community-dwelling older adults suffer from sleep problems, such as difficulty initiating or

**Data Availability Statement:** Data cannot be shared publicly due to ethical restrictions. The participants signed an informed consent form, which states that their data are exclusively available

for research institutions in an anonymized form. The raw data used in this study contains sensitive and identifying information on individuals including gender, age, and education level that could compromise the privacy of research participants. However, the data that support the findings of this study are available upon ethical approval by the local ethics committee of the Oita University Hospital. Please contact the the ethics committee of the Oita University Hospital. Email: rinrikenkyu@oita-u.ac.jp.

**Funding:** The present study was supported by Japan Agency for Medical Research and Development grants no.18he1402003 and Scientific Research (C) grants no. 19K07916. The funders had no role in study design, data collection and analysis, decision to publish, or preparation of the manuscript.

**Competing interests:** Drs Kimura, Aso, Yabuuchi, Sasaki, Nakamichi, Uesugi, Sumi, and Ms Eguchi reported that the wristband sensor was jointly developed with TDK Corporation and Oita University. There are no patents, products in development or marketed products associated with this research to declare. Dr Kimura reported receiving speaking honoraria from Takeda Pharmaceutical Co.,Ltd., Janssen Pharmaceutical, Ono Pharmaceutical Co., Ltd., Daiichi Sankyo Co., Ltd., Eisai Co.,Ltd., Dai-Nippon Sumitomo Pharma Co.,Ltd., FUJIFILM Toyama Chemical Co.,Ltd., Kyowa Hakko-Kirin Co.,Ltd., Otsuka Pharmaceutical, Co.,Ltd, and Tsumura & Co.. Dr Aso reported receiving speaking honoraria from Eisai, Co.,Ltd, Otsuka Pharmaceutical, Co.,Ltd, Kyowa Hakko-Kirin Co.,Ltd., Bayer Yakuhin, Ltd, and FP Pharma. Co. This does not alter our adherence to PLOS ONE policies on sharing data and materials.

maintaining sleep [1]. In Japan, almost 30% of older adults claim to be affected by insomnia [3]. This inevitable change is associated with an increased risk of cognitive impairment, major chronic diseases, and mortality [4–6]. Therefore, maintaining sleep quality is a key determinant of the health-related quality of life of older adults. We have previously examined an association between objectively measured lifestyle factors and global cognitive function in community-dwelling older adults [7]. Random forest regression analysis showed that daily walking steps and total sleep time were associated with Mini-Mental State Examination (MMSE) score. We suggest that daily walking steps and total sleep time are important lifestyle factors for preventing cognitive impairment in older adults. However, the association between daily walking steps and sleep parameters has not been clarified in our previous study. A growing body of evidence has shown an association between sleep and physical function or activity [8, 9]. Poor sleep quality is associated with physical inactivity and physical disability [10, 11], whereas regular physical activity is important for improving the sleep quality [12]. These results lead us to hypothesize that daily walking steps may be associated with sleep parameters in older adults. Previous studies assessed sleep using self-report questionnaires, which tend to yield poor reliability and consistency due to recall bias or misclassification, particularly among older adults. Therefore, an objective measurement of daily walking steps and sleep parameters is required to confirm their association. Wearable sensors have previously been employed to evaluate lifestyle factors, such as physical activity and sleep, in large epidemiological studies [10, 11, 13, 14]. These sensors are considered to be noninvasive, cost-effective tools to objectively measure total daily movement and sleep, and the data collected by them are not affected by recall bias. To the best of our knowledge, a few studies have reported the relationship between objectively measured moderate-to-vigorous- intensity physical activities and sleep parameters in community-dwelling older adults [15, 16]. Few studies, however, examined the association between objectively measured daily walking steps and sleep parameters in older adults. In this study, we focused on daily walking steps because engaging in exercise programs is typically difficult for older people due to their physical limitations or health conditions. Walking is a convenient and safe activity for all age groups and accounts for most of the energy expenditure among older people. Moreover, walking enhances the physical health and reduces the risk of all-cause mortality [17, 18]. Therefore, the aim of this study was to confirm whether daily walking steps are associated with the duration or quality of sleep in community-dwelling older adults using wearable sensors.

## Materials and methods

### Participants

855 community-dwelling adults [317 men, 37.1%; 538 women, 62.9%; median (interquartile range, IQR) age: 73 (69–78) years, median (IQR) educational level: 12 (11–12) years] enrolled in the community-based observational study focusing on lifestyle factors related to dementia in Usuki between August 2015 and March 2016, as described elsewhere [7]. Participants were required to wear a wristband sensor for an average of 7–14 days for every measurement period. Moreover, measurement of the lifestyle factors was repeated every three months for one year (i.e., four times per year; total study duration: 56 days) to eliminate measurement errors due to seasonal differences in lifestyle [19]. Briefly, the inclusion criteria were as follows: (1) age $\geq$ 65 years, (2) residents of Usuki, (3) physically and psychologically healthy individuals, (4) absence of dementia, and (5) ability to independently perform the activities of daily living. The exclusion criteria included a history of other neurological and psychiatric disorders (including Parkinson's disease or epilepsy), severe head trauma, alcoholism, severe cardiac failure, severe hepatic or renal dysfunction, patients undergoing treatment for cancer, and individuals

experiencing walking difficulties as a result of a stroke. All participants underwent a physical examination, a cognitive function evaluation using MMSE, and a medical interview at baseline. Height and weight were measured, and the body mass index (BMI) was calculated as weight (kg) divided by height (m$^2$). Data pertaining to demographic characteristics (including age, sex, and education level), vascular risk factors, such as hypertension, diabetes mellitus, and hypercholesterolemia, smoking status, history of alcohol consumption, diagnosis of dementia, and medication history were collected at baseline via interviews conducted by trained medical staff. History of a chronic disease was defined as a prior diagnosis of stroke, cardiac disease, hepatic or renal disease, or cancer. Assessments of vascular risk factors, such as hypertension, diabetes mellitus, and hypercholesterolemia were based on a detailed clinical and medication (antihypertensive, antidiabetic, or hypocholesterolemic medication) history. Moreover, information pertaining to a diagnosis of dementia or the administration of medication for dementia was collected from the local hospital. Information related to decreases in the activities of daily living as a result of a cognitive impairment was obtained from the participants and their closest relatives via face-to-face clinical interviews. This prospective study was conducted in accordance with the Declaration of Helsinki and was approved by the Local Ethics Committee of the Oita University Hospital (UMIN000017442). Written informed consent was obtained from all the subjects to participate in the study.

## Wearable-sensor data

All participants were asked to wear a wristband sensor (Silmee™ W20; TDK Corporation, Tokyo, Japan) on their wrist day and night, except while bathing. These wearable sensors were used to measure various lifestyle parameters, including walking steps, as well as various sleep parameters. These parameters were calculated by gathering the sensor data for each day and averaging it over the entire measurement period. The number of daily walking steps was calculated by averaging the total number of steps per day. Sleep parameters include total sleep time (TST), sleep efficiency, time awake after sleep onset (WASO), awakening time count, and naptime. TST, sleep efficiency, WASO, and awakening time count were measured from 18:00 to 05:59 on the subsequent morning. The start point was defined as the clock time associated with the beginning of the first 20 min block of sleep without movement. TST was defined as the average total number of minutes slept per day. Sleep efficiency was calculated as the percentage of TST over the time spent in bed. Nocturnal awakening was defined as 20 min of continuous movement from sleep onset to the end of sleep. WASO and awakening time count were calculated by averaging the total number of minutes awake after sleep onset and the frequency of awakening per day, respectively. Naptime was defined as rest without movement recorded on the wearable sensor from 06:00 to 17:59 on the same evening. We verified the measurement accuracy of the walking steps and sleep duration by comparing the sensor data with video observation data in healthy older participants [7]. Data are presented as "mean (standard deviation, SD)" or as "median (IQR)."

## Statistical analysis

Multiple linear regression analysis was performed to examine the association between the number of daily walking steps and different sleep parameters, such as TST, WASO, sleep efficiency, awakening time count, and naptime, after adjusting for covariates (including age, sex, educational level, BMI, vascular risk factors, alcohol consumption, and smoking status), and $p$-values of $<0.05$ were considered to indicate statistical significance. The resulting $p$-values were corrected according to the false discovery rate. All statistical analyses were conducted using IBM SPSS Statistics version 25.0 (IBM Corp., Armonk, NY, USA).

## Results

### Demographic characteristics of the study population and wristband sensor data

The median (IQR) BMI was 23.0 (21.1–25.1) kg/m$^2$ and the median (IQR) MMSE score was 29 points (27–30 points). Ever smoker was 4.2% and ever drinker was 41.4%. 429 subjects (50.2%) had a history of hypertension, 114 (13.3%) had a history of diabetes mellitus, and 281 (32.9%) had a history of hypercholesterolemia. The mean (SD) duration for which lifestyle data were collected using the wristband sensor was 31.3 days (7.1 days) per year (7.8 days on average every three months). The median (IQR) number of daily walking steps was 5,115.7 (3,395.3–7,061.4), the mean (SD) TST was 408.4 min (69.1 min), the median (IQR) WASO was 19.6 min (11.6–30.3 min), the median (IQR) sleep efficiency was 95.5% (93.1%–97.1%), the median (IQR) awakening time count was 0.46 counts (0.29–0.69 counts), and the median (IQR) naptime was 37.9 min (21.5–64.9 min). The daily walking steps and sleep parameters in our cohort were similar to those found in previous studies on Japanese adults at a comparable age [20, 21].

### Multiple linear regression analyses

Table 1 summarizes the results of the multiple linear regression analyses, showing the association between the number of daily walking steps and various sleep parameters. Daily walking steps showed a positive correlation with sleep efficiency (Fig 1A) and an inverse correlation with WASO (Fig 1B), awakening time count (Fig 1C), and naptime (Fig 1D) after adjusting for covariates and correction for the false discovery rate ($\beta = 0.098$, 95% confidence interval [CI]: 0.034 to 0.162, $p = 0.003$; $\beta = -0.107$, 95% CI: $-0.172$ to $-0.043$, $p = 0.001$; $\beta = -0.105$, 95% CI: $-0.17$ to $-0.04$, $p = 0.002$; and $\beta = -0.31$, 95% CI: $-0.371$ to $-0.249$, $p < 0.001$, respectively). However, the correlation between the number of daily walking steps and TST was attenuated to a nonsignificant trend after adjusting for covariates ($\beta = -0.001$, 95% CI: $-0.066$ to 0.065, $p = 0.99$).

## Discussion

We herein examined the association between daily walking steps and sleep parameters in community-dwelling older adults. Several studies have examined the association between moderate-to-vigorous- intensity physical activities and sleep parameters using self-report questionnaires. We have previously examined an association between lifestyle factors and global cognitive function and suggested that daily walking steps and total sleep time were important for preventing cognitive impairment [7]. However, the association between daily

**Table 1. Multiple regression model showing the association between the daily walking steps and sleep parameters.**

|  | Walking steps | |
|---|---|---|
|  | $\beta$ (95% CI) | $p$-value |
| TST | −0.001 (−0.066, 0.065) | 0.99 |
| Sleep efficiency | 0.098 (0.034, 0.162) | 0.003* |
| WASO | −0.107 (−0.172, −0.043) | 0.001* |
| Awakening time count | −0.105 (−0.17, −0.04) | 0.002* |
| Naptime | −0.31 (−0.371, −0.249) | <0.001* |

TST, total sleep time; WASO, time awake after sleep onset; CI, confidence interval.

*$p < 0.05$.

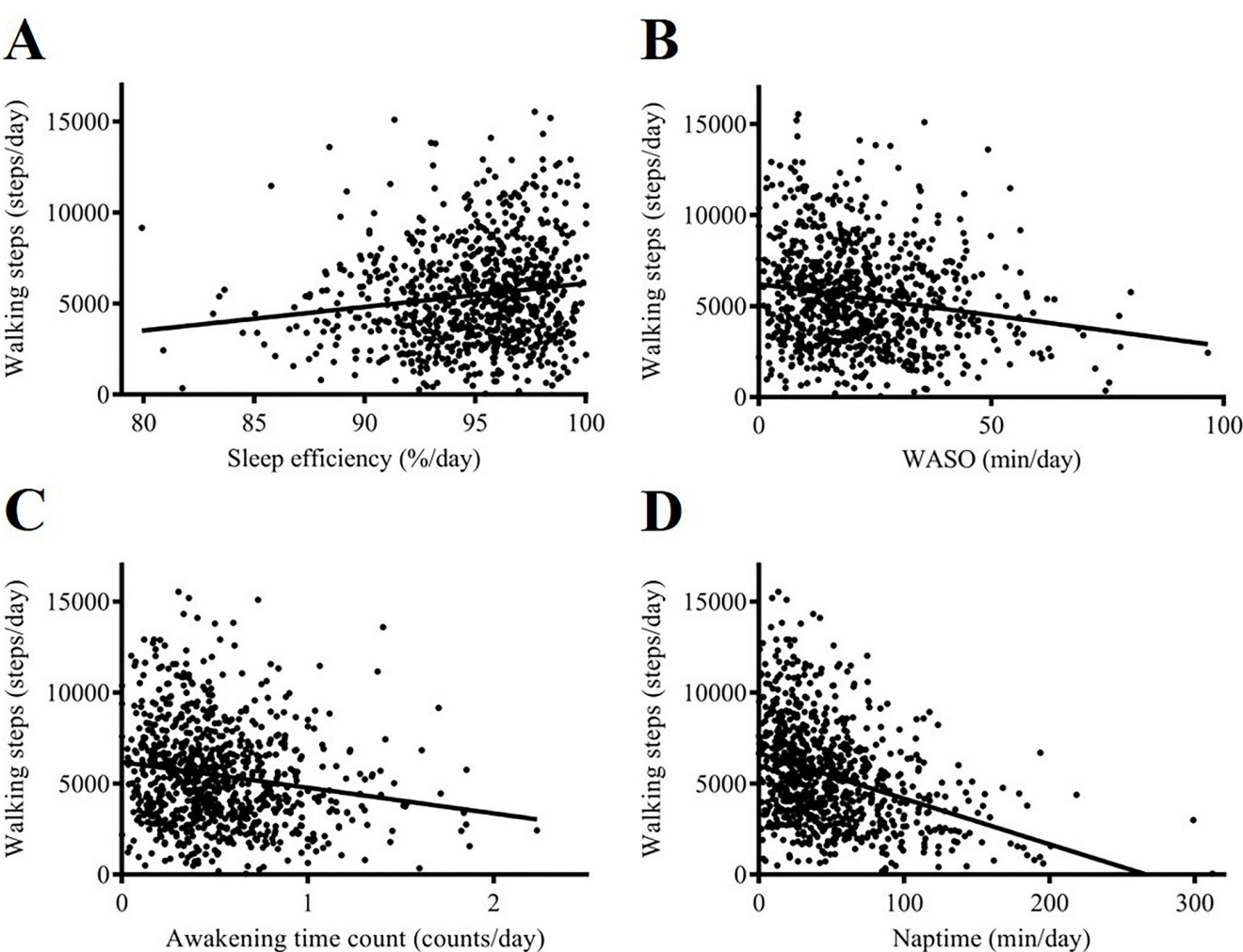

**Fig 1. Association between daily walking steps and sleep parameters.** Daily walking steps showed a positive correlation with sleep efficiency (A) and an inverse correlation with WASO (B), awakening time count (C), and naptime (D). WASO, time awake after sleep onset.

walking steps and sleep parameters has not examined in this cohort. To the best of our knowledge, this is the first study clarifying the association between objectively measured daily walking steps and sleep parameters in older adults using a wearable sensor. The present study provided novel and interesting insights for devising public health strategies to improve the quality of sleep. Daily walking steps showed a positive correlation with sleep efficiency and an inverse correlation with WASO, awakening time count, and naptime after adjusting for covariates and correcting for the false discovery rate. This result indicates an association between the walking steps and sleep quality of older adults. This study has several strengths; this was a large-scale study of community-dwelling older adults, and it included objective measurements of daily walking steps and sleep parameters.

The most interesting finding in this study was that the increase in the daily walking steps was associated with greater sleep efficiency, lower frequency of awakening after sleep onset, and shorter WASO and naptime. Several studies have examined the association between physical function or activity and sleep parameters in older adults [10, 11, 14, 15, 22–26]. Despite the varied study designs and methods used for sleep assessment, previous studies have revealed a robust association between physical function or activity and sleep parameters. For example,

cross-sectional studies employing self-reported questionnaires showed an association among long sleep duration, poor sleep quality, and decreased physical function [2, 15, 22–25], whereas other studies employing actigraphy showed an association among short or long sleep duration, poor sleep quality, and decreased physical function [10, 11]. A longitudinal study employing self-reported measures showed an association between long sleep duration and decreased physical function [26], and another study employing actigraphy showed an association between poor sleep quality (characterized by greater WASO and lower sleep efficiency) and decreased physical function [14]. However, the present study differs from previous studies in that it included an objective measurement of both physical activity and sleep parameters using a wearable sensor, which eliminated the risk of recall bias or misclassification. Only one study reported the temporal and bidirectional association between objectively measured sleep parameters and physical activity indices in older adults [15]. This previous study, however, included a small number of healthy women ($n$ = 143, mean age: 73 years) and assessed a moderate-to-vigorous-intensity physical activity. In contrast, we measured daily walking in a large number of older adults. Walking is a convenient and safe activity for older adults. Moreover, interventional studies have shown that walking is better than vigorous physical activity with respect to achieving a good sleep quality [27–29]. Therefore, we suggest that increased daily walking steps may be associated with a good sleep quality. Although the mechanism underlying the association between physical activity and sleep is not well understood, physical activity can reduce systemic inflammation and improve body composition or psychological well-being [30].

Our results showed no significant correlation between daily walking steps and sleep duration. Previous studies employing actigraphy have shown contradictory results regarding association physical function and sleep duration [10, 14]. One study showed the association between decreased physical function and short or long sleep duration [10], whereas the other study showed no significant association between physical function and sleep duration [14]. Although it remains unclear whether daily walking steps is more closely associated with sleep quality compared to sleep duration, a previous interventional study reported that walking could improve sleep quality rather than sleep duration [29].

Some limitations in our study should be considered while interpreting the results. First, the cross-sectional design of the study does not permit any causal inferences. Second, we did not assess other type of physical activity, including moderate- or vigorous- intensity physical activity. Further studies are needed to determine which type of physical activity is associated with sleep quality or duration. Third, we collated clinical data to define the presence or absence of dementia; however, patients with possible dementia may not have been completely excluded from the study. Although all participants were decided to be physically and psychologically healthy by physical examination medical interview at baseline, the participants with chronic disease could not be excluded completely from participating in the current study.

In conclusion, to the best of our knowledge, this is the first study demonstrating an association between objectively measured daily walking steps and sleep quality. Increased daily walking steps are associated with greater sleep efficiency, lower frequency of awakening time count, and shorter WASO and naptime. The current results may contribute to the development of new evidence-based interventions for improving sleep quality in community-dwelling older adults.

## Acknowledgments

We gratefully acknowledge the assistance of Usuki city employees for their efforts in recruiting participants. We thank the TDK Corporation for the development of the wearable sensor,

Suzuki Co. Ltd. for their assistance with data collection, and HCL Technologies, Confidential, and Fusa Matsuzaki for database construction and data analysis. We received generous assistance from Kaori Hirano, Megumi Ogata, Mai Kishigami, and Oita University students with the physical measurement and interview procedures.

## Author Contributions

**Conceptualization:** Noriyuki Kimura, Etsuro Matsubara.

**Data curation:** Noriyuki Kimura, Yasuhiro Aso, Kenichi Yabuuchi.

**Formal analysis:** Yasuhiro Aso.

**Funding acquisition:** Etsuro Matsubara.

**Investigation:** Noriyuki Kimura, Kenichi Yabuuchi.

**Supervision:** Etsuro Matsubara.

**Writing – original draft:** Noriyuki Kimura.

**Writing – review & editing:** Etsuro Matsubara.

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
