## [Decision Letter · Decision Letter 0]

21 Oct 2020

PONE-D-20-10392

Association between objectively measured walking steps and sleep in community-dwelling older adults: A prospective cohort study

PLOS ONE

Dear Dr. Kimura,

Thank you for submitting your manuscript to PLOS ONE. After careful consideration, we feel that it has merit but does not fully meet PLOS ONE’s publication criteria as it currently stands. Therefore, we invite you to submit a revised version of the manuscript that addresses all the points raised during the review process.

We look forward to receiving your revised manuscript.

Kind regards,

Gianluigi Forloni

Academic Editor

PLOS ONE

Journal Requirements:

"We gratefully acknowledge the assistance of Usuki city employees for their efforts in recruiting

238 participants. We thank the TDK Corporation for the development of the wearable sensor, Suzuki

239 Co. Ltd. for their assistance with data collection, and HCL Technologies, Confidential, and Fusa

240 Matsuzaki for database construction and data analysis. We received generous assistance from

241 Kaori Hirano, Megumi Ogata, Mai Kishigami, and Oita University students with the physical

242 measurement and interview procedures."

i) We note that you have provided funding information that is not currently declared in your Funding Statement. However, funding information should not appear in the Acknowledgments section or other areas of your manuscript. We will only publish funding information present in the Funding Statement section of the online submission form.

ii) Please remove any funding-related text from the manuscript and let us know how you would like to update your Funding Statement. Currently, your Funding Statement reads as follows:

" This research was supported by Japan Agency for Medical Research and Development [Grant Number 18he1402003]. The founders had no role in design and conduct of the study; collection, management, analysis, and interpretation of the data; preparation, review, or approval of the manuscript; and decision to submit the manuscript for publication.".

iii) Additionally, because some of your funding information pertains to commercial funding we ask you to provide an updated Competing Interests statement, declaring all sources of commercial funding. 

iv) In your Competing Interests statement, please confirm that your commercial funding does not alter your adherence to PLOS ONE Editorial policies and criteria by including the following statement: "This does not alter our adherence to PLOS ONE policies on sharing data and materials.” as detailed online in our guide for authors  http://journals.plos.org/plosone/s/competing-interests.  If this statement is not true and your adherence to PLOS policies on sharing data and materials is altered, please explain how.

 * Please include the updated Competing Interests Statement and Funding Statement in your cover letter. We will change the online submission form on your behalf.

Reviewers' comments:

Reviewer's Responses to Questions

**Comments to the Author**

1. Is the manuscript technically sound, and do the data support the conclusions?

Reviewer #1: Yes

2. Has the statistical analysis been performed appropriately and rigorously? 

Reviewer #1: Yes

3. Have the authors made all data underlying the findings in their manuscript fully available?

Reviewer #1: Yes

4. Is the manuscript presented in an intelligible fashion and written in standard English?

Reviewer #1: Yes

5. Review Comments to the Author

Reviewer #1: Abstract:

The introduction in the abstract makes the reader think that only steps from walking around the house and from fidgeting will be taken into consideration for this study. Since this is not the case, this should be removed.

Introduction:

Page 4, line 63- the authors mention that only one study has assessed the relationship between MVPA and sleep parameters, however, there have been other studies conducted on the subject matter -Gabriel, Kelley Pettee, et al. "Bidirectional associations of accelerometer-determined sedentary behavior and physical activity with reported time in bed: Women's Health Study." Sleep health 3.1 (2017): 49-55.

The authors state that they will focus on daily walking steps since it is a measure of low-intensity activity-is there a citation for this? How are the authors able to discern whether steps taken belonged in the light or moderate-to-vigorous intensity category? The authors may consider not talking about intensity since it is not reporting OR they may consider reporting time spent in light, moderate and vigorous intensity.

Methods:

Page 5, Line 83- how did the authors ensure “physical and psychological health”? What if they had a chronic disease?

Discussion: Page 10, line 191, citation missing for “several studies” which ones?

There are several grammatical errors throughout the manuscript.

Keywords: Community is misspelled

Page 3, line 54- Remove “almost”

Page 9, line 179, nor should be “not”

6. PLOS authors have the option to publish the peer review history of their article (what does this mean?). If published, this will include your full peer review and any attached files.

Reviewer #1: No

---

## [Author Response · Author response to Decision Letter 0]

5 Nov 2020

We would like to take this opportunity to express our sincere thanks to the reviewer who identified areas of the manuscript that needed corrections or modification. Based on the instructions provided in the decision letter and comments provided by the reviewers, we have revised the manuscript by modifying the relevant sections in the manuscript. Also appended below are point-by-point responses to the comments raised by the reviewers. We hope that our revisions along with our responses address reviewer’s concerns and that our revised manuscript is now suitable for publication in PLOS ONE.

Reviewer 1

We thank Reviewer #1 for the critical comments and useful suggestions, which have substantially helped us improve our manuscript. As indicated in our responses below, we have considered each reviewer comment and suggestion and have revised the manuscript accordingly. We hope that our responses and revisions are appropriate and that our revised manuscript is considered for publication.

1. “Reviewer’s comment” 

Abstract: The introduction in the abstract makes the reader think that only steps from walking around the house and from fidgeting will be taken into consideration for this study. Since this is not the case, this should be removed.

 “Author’s response”

We agreed with reviewer’s comment. We have removed this sentence and added the sentence in the Abstract section, as follows:

P 2, line 15- 16

There is increasing evidence that physical activity is associated with sleep quality.

2. “Reviewer’s comment” 

Introduction: Page 4, line 63- the authors mention that only one study has assessed the relationship between MVPA and sleep parameters, however, there have been other studies conducted on the subject matter -Gabriel, Kelley Pettee, et al. "Bidirectional associations of accelerometer-determined sedentary behavior and physical activity with reported time in bed: Women's Health Study." Sleep health 3.1 (2017): 49-55.

“Author’s response”

We agreed with reviewer’s comment. We have revised the relevant sentences in the Methods section, as follows:

P 4, line 60- 63

To the best of our knowledge, a few studies have reported the relationship between objectively measured moderate-to-vigorous- intensity physical activities and sleep parameters in community-dwelling older adults [15, 16].

3. “Reviewer’s comment” 

The authors state that they will focus on daily walking steps since it is a measure of low-intensity activity-is there a citation for this? How are the authors able to discern whether steps taken belonged in the light or moderate-to-vigorous intensity category? The authors may consider not talking about intensity since it is not reporting OR they may consider reporting time spent in light, moderate and vigorous intensity.

 “Author’s response”

Several previous studies reported that walking may be categorized into light-intensity physical activity, as follows:

“Whether the association includes low-intensity activity such as regular walking is not known.” Abbott RD, et al. JAMA. 2004;292:1447-1453.

 “Women may　spend more time doing low and lifestyle intensity activities,　such as walking, household chores, and gardening.”　Lohne-Seiler et al. BMC Public Health 2014, 14:284. 

“Recent studies have　shown that light-intensity PA (LPA; e.g., housework, gardening,　and casual walking) and sedentary behavior (SB;e.g., television viewing, computer use, workplace sitting,　and sitting in automobile) are also related to health of older　adults.”　Yasunaga A, et al. Health and Quality of Life Outcomes. 2018;16:240. 

“These activities can include non-exercise leisure-time and life-style activities (e.g.　walking, gardening, etc.) and instrumental activities of daily living (IADLs) (e.g. shopping, housework, etc.), which are typically in the low-intensity range. Research using self-report measures of walking activity indicates that these non-exercise physical activities may be associated with cognitive health benefits.” Varma VR, et al. Hippocampus. 2015; 25: 605–615.

Therefore, we classified walking steps into light-intensity activity. However, we agreed with reviewer’s comment that walking steps include the light or moderate-to-vigorous intensity activity. Moreover, in the present study, we have not investigated the intensity of physical activity. Therefore, we have revised the relevant sentences in the Introduction and Discussion sections, as follows:

P 4, line 64- 68

In this study, we focused on daily walking steps because engaging in exercise programs is typically difficult for older people because of their physical limitations or health conditions. Walking is a convenient and safe activity for all age groups and accounts for most of the energy expenditure among older people. 

P 11, line 208- 209

In contrast, we measured daily walking in a large number of older adults.

4. “Reviewer’s comment” 

Methods:

Page 5, Line 83- how did the authors ensure “physical and psychological health”? What if they had a chronic disease? 

“Author’s response”

We agreed with reviewer’s comment. We have added the relevant sentences in the Discussion section, as follows:

P 11, line 229- P 12, line 231

Although all participants were decided to be physically and psychologically healthy by physical examination medical interview at baseline, the participants with chronic disease could not be excluded completely from participating in the current study. 

5. “Reviewer’s comment” 

Discussion: Page 10, line 191, citation missing for “several studies” which ones?

“Author’s response”

We agreed with reviewer’s comment. We have revised the relevant sentences in the Discussion section, as follows:

P 10, line 192

Several studies have examined the association between physical function or activity and sleep parameters in older adults [10, 11, 14, 15, 22–26]. 

6. “Reviewer’s comment” 

There are several grammatical errors throughout the manuscript.

Keywords: Community is misspelled

Page 3, line 54- Remove “almost”

Page 9, line 179, nor should be “not”Discussion:

“Author’s response”

We agreed with reviewer’s comment. We have corrected grammatical errors throughout the manuscript.

---

## [Editor Report · Decision Letter 1]

1 Dec 2020

Association between objectively measured walking steps and sleep in community-dwelling older adults: A prospective cohort study

PONE-D-20-10392R1

Dear Dr. Kimura,

We’re pleased to inform you that your manuscript has been judged scientifically suitable for publication and will be formally accepted for publication once it meets all outstanding technical requirements.

Kind regards,

Gianluigi Forloni

Academic Editor

PLOS ONE
---

## [Editor Report · Acceptance letter]

4 Dec 2020

PONE-D-20-10392R1 

Association between objectively measured walking steps and sleep in community-dwelling older adults: A prospective cohort study 

Dear Dr. Kimura:

I'm pleased to inform you that your manuscript has been deemed suitable for publication in PLOS ONE. Congratulations! Your manuscript is now with our production department. 

Kind regards, 

on behalf of

Dr. Gianluigi Forloni 

Academic Editor

PLOS ONE